# Anthraquinone-Polyaniline-Integrated Textile Platforms for In Situ Electrochemical Production of Hydrogen Peroxide for Microbial Deactivation

**DOI:** 10.3390/polym15132859

**Published:** 2023-06-28

**Authors:** Samuel M. Mugo, Weihao Lu, Scott Robertson

**Affiliations:** Physical Sciences Department, MacEwan University, 10700-104 Avenue, Edmonton, AB T5J 4S2, Canada

**Keywords:** electrocatalytic H_2_O_2_ production, *E. coli* deactivation, anthraquinone-polyaniline-integrated antibacterial textile patch

## Abstract

Hydrogen peroxide (H_2_O_2_) is a versatile and effective disinfectant against common pathogenic bacteria such as *Escherichia coli* (*E. coli*). Electrochemical H_2_O_2_ generation has been studied in the past, but a lack of studies exists on miniaturized electrochemical platforms for the on-demand synthesis of H_2_O_2_ for antibacterial applications. In this article, a chemically modified cotton textile platform capable of in situ H_2_O_2_ production is demonstrated for *E. coli* deactivation. The cotton textile was modified by layer-by-layer coating with conductive carbon nanotubes/cellulose nanocrystals (CNT/CNC) and a polymer of polyaniline (PANI) decorated with anthraquinone (AQ), designated as the AQ@PANI@CNT/CNC@textile antibacterial patch. The AQ@PANI@CNT/CNC@textile antibacterial textile patch H_2_O_2_ production capabilities were evaluated using both electrochemical and colorimetric methods. The AQ@PANI@CNT/CNC@textile antibacterial patch electrochemically produced H_2_O_2_ concentrations up to 209 ± 25 µM over a 40 min period and displayed a log reduction of 3.32 for *E. coli* over a period of 2 h. The AQ@PANI@CNT/CNC@textile antibacterial patch offers promise for use as a self-disinfecting pathogen control platform.

## 1. Introduction

Hydrogen peroxide (H_2_O_2_) is an environmentally friendly oxidant used in a wide range of applications, including energy production, chemical synthesis, wastewater treatment, and bioremediation [1,2]. Additionally, it may act as a bacterial and viral pathogenic disinfectant [3,4]. As a disinfectant, H_2_O_2_ causes the oxidation of proteins, cleavage of nucleic acids, and dissociation of the 70S ribosomal subunits and denatures a pathogens cell wall [4]. Due to the 2020 COVID-19 pandemic, there is renewed interest in the use of H_2_O_2_ for the deactivation of pathogens. For example, some studies have examined the usage of H_2_O_2_ for the sterilization and reuse of personal protective equipment (PPE) to address global shortages and for environmental conservation motivations [5,6,7,8]. However, sterilized surgical masks and respirators must adhere to strict FDA guidelines of having a log reduction of ≥3 of both 2 g positive and 2 g negative vegetative bacteria [8]. Recent studies report that 0.5% H_2_O_2_ cooling aerosol sprays were found to be highly effective in reducing the viral coronavirus load on used face masks [9,10]. Because of its inherent disinfectant capabilities_,_ the development of new methods for on-demand H_2_O_2_ production is an important area of research.

The standard method of H_2_O_2_ production includes a complex anthraquinone (AQ)-based process [11]. In this process, AQ undergoes hydrogenation and oxidation in the presence of a nickel or palladium catalyst, yielding H_2_O_2_ as a byproduct, which is then distilled and concentrated [1,11]. However, this process requires a high energy input, produces large amounts of waste, presents the risk of explosions, and is not suitable for producing small, on-demand quantities of H_2_O_2_ for integration into PPE [11]. To overcome these drawbacks, electrochemical methods for H_2_O_2_ production have been explored. For instance, electrochemical technologies have utilized metal catalysts, such as platinum, to produce H_2_O_2_ [1]. Additionally, Xue et al. developed a carbon- and nitrogen-doped TiO_2_ electrode capable of converting H_2_O into H_2_O_2_ at a rate of 0.29 µmol·L^−1^·cm^−2^·h^−1^, with a faradaic efficiency of 8% [2]. In a different approach, an electrocatalyst comprising polyaniline (PANI) and AQ has been developed to catalyze the reduction of O_2_ into H_2_O_2_ [12]. Utilizing an electrosynthesis flow cell, H_2_O_2_ was generated at a rate of 1.80 mol·g^−1^·hr^−1^, with a faradaic efficiency of 95.8%, making their platform a promising new technology [12].

Both PANI and AQ are capable of generating H_2_O_2_ on the cathode when immersed in an electrochemical cell [12,13]. When PANI is oxidized from its leucoemeraldine form to its emeraldine form, it electrocatalyzes the reduction of oxygen via a 2e^−^ oxygen reduction reaction (ORR) transition, shown in Equations (1) and (2) [12,13].
(PANI)_ox_ + 2*n*H^+^ + 2*n*e^−^ → (PANI)_red_,(1)
(PANI)_red_ + *n*O_2_ → (PANI)_ox_ + *n*H_2_O_2_,(2)

A 2e^−^ ORR may also be catalyzed via AQ, and this reaction is a more kinetically favorable pathway than that afforded via PANI [14,15]. At the cathode, AQ reduces to its semiquinone form, which then catalyzes the reduction of oxygen to the superoxide anion radical (O_2_^·−^), which becomes protonated to form H_2_O_2_ (Figure 1) [14,15]. It is hypothesized that the integration of AQ into the PANI network will likely dramatically enhance the efficiency of H_2_O_2_ production [14,15].

While effective, metal electrocatalysts are susceptible to anodic degradation and may require regeneration between uses [2]. Furthermore, the creation and integration of a miniaturized electrochemical H_2_O_2_-producing device for bacterial and viral disinfection remain a challenge. This study demonstrates a layer-by-layer (LbL) assembly of a cotton textile substrate integrated with layers of conductive carbon nanotube (CNT)/cellulose nanocrystal (CNC) composite and PANI decorated with AQ (AQ@PANI@CNT/CNC@textile). The AQ@PANI@CNT/CNC@textile platform has been successfully tested for electrochemical production of H_2_O_2_ for *E. coli* deactivation. The H_2_O_2_ production capabilities of the AQ@PANI@CNT/CNC@textile antibacterial patch were quantified using cyclic voltammetry (CV) and colorimetric methods. The H_2_O_2-_generating AQ@PANI@CNT/CNC@textile holds promise for potential integration into masks or other wearable devices for in situ disinfection.

## 2. Experimental

### 2.1. Materials

Ammonium peroxydisulfate (APS), 3-(trimethoxysilyl)propyl methacrylate (TPM), methanol, silver nanoparticles, 2,6-diaminoanthraquinone, sodium hypochlorite, dimethylsulfoxide (DMSO), LB broth with agar (Lennox), potassium ferricyanide, sodium acetate trihydrate, 4-aminoantipyrine, phenol, acetic acid, horseradish peroxidase, *E. coli* K-12 (Z-), tryptic soy agar, saline solution, and barium chloride were procured from Sigma Aldrich, Oakville, ON, Canada. Sulfuric acid, 30% H_2_O_2_, aniline, dipotassium phosphate, and monopotassium phosphate were purchased from Fisher Scientific, Denver, CO, USA. Carboxylic acid-functionalized multiwalled carbon nanotubes (CNT) (OD: 4–6 nm. 98%) were bought from TimesNano, China. Cellulose nanocrystals (CNC) were donated by Alberta Innovates, Canada. Stainless steel hypodermic needles (0.7 mm × 40 mm, inner diameter 0.5 ± 0.1 mm) and textile bandages were bought from a local pharmacy in Edmonton, Alberta. Copper tape was purchased from 3M (Milton, ON, Canada). All reagents were of analytical reagent grade. All aqueous solutions were prepared using 18.2 MΩ deionized (DI) Milli-Q water.

### 2.2. CNT/CNC H_2_O_2_ Sensor Electrode Fabrication

The CNT/CNC sensor electrode for H_2_O_2_ detection and the CNT/CNC/Ag (Ag/AgCl) reference needle were fabricated as described previously [16,17]. Briefly, stainless steel hypodermic needles were oxidized via immersion in piranha solution consisting of concentrated H_2_SO_4_ and 30% H_2_O_2_ in a 1:1 (*v*/*v*) ratio for 4 h, then washed with DI water and air dried. The oxidized needles were then silylated via immersion in a solution comprising 3-(trimethoxysilyl) propyl methacrylate (TPM)/DI water/methanol (2:1:8 *v*/*v*/*v*) for 4 hrs, then washed with DI water and air dried. To fabricate the CNT/CNC sensor electrode, the silylated needles were infused with a homogenous suspension comprising 1 mg·mL^−1^ CNT/4 mg·mL^−1^ CNC at a flow rate of 15 µL·min^−1^ at 80 °C. Homogenous CNT/CNC suspensions were first prepared by sonicating 1 mg·mL^−1^ CNT/4 mg·mL^−1^ CNC in DI water for 3 h [17]. To fabricate the in-house Ag/AgCl reference needle, silylated needles were infused with a 1 mg·mL^−1^ CNT/4 mg·mL^−1^ CNC/5 mg·mL^−1^ Ag nanoparticle suspension (in DI water) at a flow rate of 15 µL·min^−1^. The needles were then immersed in 2 mL of sodium hypochlorite bleach overnight at room temperature and rinsed with DI water.

### 2.3. AQ@PANI@CNT/CNC@Textile Antibacterial Patch Fabrication

To fabricate the antibacterial patch, an 8 × 8 cm cotton textile obtained from a local store was drop cast with 8 mL of a 1 mg·mL^−1^ CNT/4 mg·mL^−1^ CNC suspension and dried on a hot plate at 50 °C. The PANI microparticles were produced by mixing 50 mL of 0.2 M aniline (in 1 M H_2_SO_4_) and 50 mL of 0.25 M APS initiator in an ice bath at 4 °C. The resulting PANI microparticle suspension was then centrifuged (4000 rpm, 15 min) and rinsed with DI water three times before being dried in a dehydrator at 70 °C. A 20 mg·mL^−1^ solution comprising dried PANI microparticles (in DI water) was sonicated for 3 h to create a homogenous suspension. A total of 8 mL of the 20 mg·mL^−1^ PANI suspension was drop cast onto the CNT/CNC@textile patch and air dried. Lastly, the PANI@CNT/CNC@textile patch was cut into 4 × 4 cm squares. To integrate AQ into the PANI@CNT/CNC@textile patches, 1.5 mL of 0.5 M 2,6-diaminoanthraquinone (in DMSO) was drop cast onto the patch surface and air dried to yield the AQ@PANI@CNT/CNC@textile antibacterial patches. The AQ was similarly drop cast onto the CNT/CNC@textile patches to yield AQ@CNT/CNC@textile patches. After air drying, the patches were gently rinsed in a beaker of DI water to remove any excess AQ. A 0.3 × 1.0 cm strip of copper tape was wrapped around one corner of each patch to serve as an electrical connection.

### 2.4. Characterization of AQ@PANI@CNT/CNC@Textile Antibacterial Patch

Fourier transform infrared spectroscopy (FTIR), scanning electron microscopy (SEM), electrochemical impedance spectroscopy (EIS), voltammetric, and Raman spectroscopy techniques were used to characterize the AQ@PANI@CNT/CNC@textile antibacterial patch. FTIR spectra were recorded using a Bruker Tensor 27 FTIR instrument fitted with diamond attenuated total reflectance (ATR). SEM images were acquired from a Zeiss Sigma 300 VP field emission SEM. EIS was performed by immersing different patches in 25 mM potassium ferricyanide (in 0.1 M KCl) as a standard redox probe in a three-electrode electrochemical cell, with an in-house fabricated Ag/AgCl reference needle, and a platinum counter electrode. EIS data were acquired in the frequency range of 20.0 Hz–1 MHz at 6.00 mV of sinusoidal amplitude. CV was used to determine the electroactive surface area of the different patches. The CV was performed by immersing each patch in 25 mM potassium ferricyanide (in 0.1 M KCl) in a three-electrode electrochemical cell, using an Ag/AgCl reference needle and a platinum counter electrode. A Palmsens 4 potentiostat equipped with PSTrace software was used for all electrochemical measurements.

### 2.5. In Situ Electrochemical Generation of H_2_O_2_, and Electrochemical and Colorimetric Quantification

To electrochemically generate H_2_O_2_, antibacterial patches were immersed in 10 mL of 0.1 M phosphate buffer (pH 7.0) electrolyte in a three-electrode cell comprising a platinum counter electrode and an Ag/AgCl reference needle for CV application (−1.0 to 1.0 V, 0.25 V·s^−1^).

The H_2_O_2_ was electrochemically detected using CV (−1.0 to 1.0 V, 0.1 V·s^−1^) acquired from a CNT/CNC needle sensor running separately within the H_2_O_2_ generation cell. The H_2_O_2_ detection cell comprised a three-electrode system, with a platinum wire and Ag/AgCl reference needle acting as the counter and reference electrodes, respectively. Faradaic capacitance was determined from the CVs acquired using the CNT/CNC needle sensor and was calculated by averaging the cathodic current within the − 0.50 to 0 V voltage range and dividing it by the scan rate. The ∆capacitance was determined by subtracting the capacitance generated from the blank from that generated from the sample and dividing it by capacitance of the blank.

A colorimetric method was also used to detect H_2_O_2_ generation. Briefly, the colorimetric reagent was prepared by combining 0.3681 g of sodium acetate trihydrate, 0.0102 g of 4-aminoantipyrine, 0.0047 g of phenol, 29.4 µL of glacial acetic acid, and D.I water to make 50 mL of solution. In a cuvette, 2 mL of the colorimetric reagent was mixed with 10 µL of 1.1 mg/mL horseradish peroxidase dissolved in D.I water, 500 µL of electrolyte, and brought to a volume of 3 mL using 490 µL of DI water [18]. The absorbance at 505 nm was measured in triplicate using a LabQuest spectrophotometer.

### 2.6. E. coli Deactivation Test Using the Antibacterial Patches

Different antibacterial patch designs were tested for their ability to deactivate *E. coli. E. coli* K-12 (Z-) cells were streaked on tryptic soy agar (TSA) and incubated at 37 °C for 24 h. Single *E. coli* colonies were suspended in 0.85% (w/v) sterile saline, and the turbidity was adjusted to match a 0.5 McFarland standard, resulting in a bacterial suspension of approximately 10^8^ cells·mL^−1^. The bacterial suspension was then diluted by combining 2.5 mL suspension with 2.5 mL of sterile saline. To incorporate *E. coli* into the antibacterial patch network for the deactivation test, 500 µL of dilute bacterial suspension was drop-casted onto the patch surface. CV (−1.0 to 1.0 V, 0.25 V·s^−1^) was subsequently performed via immersion in 10 mL of 0.1 M phosphate buffer under the same conditions used for electrochemical H_2_O_2_ generation. A total of 10 µL of phosphate buffer was drawn at various points in time (0, 5, 11, 16, 21, 27, 32, 60, 80, and 120 min), suspended in 990 µL of sterile water, and then serially diluted in a 1:9 ratio five times. The final diluted suspension was then cultured on LB agar and incubated at 37 °C for 16 h. Bacterial growth was visually evaluated by counting the number of colonies formed on the LB agar plate after the 16 h incubation period.

## 3. Results and Discussion

### 3.1. Morphological Characterization of Antibacterial Textile Patches

The surface morphologies of the various patches prior to and after *E. coli* integration were examined using SEM. The CNT/CNC@textile patch, shown in Figure 2a, shows evidence of the nanoporous and stable fiber network characteristic of the overlapping nanofiber structure of the cotton textile and CNT/CNC composite. The addition of PANI to the CNT/CNC@textile enhances patch rigidity. As shown in Figure 2b, the PANI microparticles are distributed on the CNT/CNC and textile fiber network. Figure 2c,d show the AQ@CNT/CNC@textile and the AQ@PANI@CNT/CNC@textile patches following the incorporation of *E. coli* cells, respectively. AQ is visualized as tiny crystals coated on the surface of the CNT/CNC fiber network, while *E. coli* cells appear as rounded cylinders (Figure 2c,d). Visually, the density of *E. coli* cells was higher in the AQ@PANI@CNT/CNC@textile patch, which provides evidence that the integration of the porous PANI coating can trap more bacteria.

To confirm the success of the LbL patch assembly, FTIR was acquired at various stages during AQ@PANI@CNT/CNC@textile antibacterial patch fabrication. FTIR spectra from the CNT/CNC@textile patch showed peaks at 1091, 1714, 2910, and 3423 cm^−1^ (Figure 3), which can be attributed to C-O stretching, C=O stretching, and C-H stretching of the carboxy groups attached to the walls of the functionalized CNTs, and OH stretching of carboxylic acid group, respectively [19,20]. The addition of the PANI layer to the CNT/CNC@textile patch results in two new peaks at 1495 and 1583 cm^−1^ (Figure 3), which represent the C=C stretching vibrations of the benzenoid and quinoid rings of PANI [21]. Spectra acquired from the AQ@PANI@CNT/CNC@textile antibacterial patch show an additional peak at 1567 cm^−1^, which can be attributed to the anthraquinone ring, as well as peaks around 3330, and 3417 cm^−1^ (Figure 3), which are indicative of the symmetric and asymmetric stretching vibrations of NH_2_ group [22].

### 3.2. Electrochemical Characterization of Antibacterial Patches

Different patches were characterized via EIS using 25 mM potassium ferricyanide (in 0.1 M KCl) as the standard redox probe. The Nyquist plots from EIS analysis and associated circuit fittings are shown in Figure 4a and Appendix A, respectively. A low electron transfer resistance (R_ct_) is desirable for high patch conductivity and effective signal transduction for H_2_O_2_ generation [16]. The R_ct_ values determined from the circuit fittings (Appendix A) for the CNT/CNC@textile, PANI@CNT/CNC@textile, AQ@CNT/CNC@textile, and AQ@PANI@CNT/CNC@textile patches are shown in Table 1. The integration of either PANI or AQ to the CNT/CNC@textile patches greatly reduced the R_ct_, indicative of increased electrical conductivity. When combined, the AQ@PANI@CNT/CNC@textile antibacterial patch demonstrates a reduced R_ct_ compared to the AQ@CNT/CNC@textile patch, owing to the enhanced conductivity afforded to it via the PANI integration. A lower R_ct_ is advantageous for the AQ@PANI@CNT/CNC@textile patches, as it likely results in more effective H_2_O_2_ production via enhanced electrical signal transduction.

The electroactive surface areas of the different patches were determined by running CV of the redox couple 25 mM potassium ferricyanide (in 0.1 M KCl) at different scan rates (0.025, 0.03, 0.04, 0.05, 0.1, 0.2 and 0.3 V·s^−1^) and invoking the Randles–Sevcik equation (Figure 4b) [23]. The electroactive surface areas of the CNT/CNC@textile, PANI@CNT/CNC@textile, AQ@CNT/CNC@textile, and AQ@PANI@CNT/CNC@textile patches are shown in Table 1. The addition of the PANI layer resulted in an increase in the electroactive surface area of the textile patches relative to their non-PANI-integrated counterparts. A high electroactive surface area is desirable for an effective electrochemical platform and lends well to affording a higher capacity for bacterial adsorption. The addition of AQ only slightly reduced the electroactive surface area of the PANI@CNT/CNC@textile patches.

### 3.3. Electrochemical and Colorimetric Quantification of Antibacterial Textile Patch H_2_O_2_ Production

Two three-electrode cells were utilized to produce and quantify the H_2_O_2_ generated from various chemically modified patches. In the H_2_O_2_ generation cell, the patches acted as the working electrode for CV application in phosphate buffer. For the H_2_O_2_ detection system, a CNT/CNC sensor electrode was used as the working electrode for CV acquisition in the same electrochemical cell using a different potentiostat [17]. Representative CVs taken from CNT/CNC@textile, PANI@CNT/CNC@textile, and AQ@PANI@CNT/CNC@textile patches acquired during the electrochemical catalysis of H_2_O_2_ are shown in Appendix A.

For the capacitive quantification of H_2_O_2_, a calibration curve in the range of 0–400 µM H_2_O_2_ was used (Appendix A), where the ∆capacitance was determined using the cathodic peak current found within the −0.5 to 0 V range. Evidently, the CNT/CNC sensor electrode used for H_2_O_2_ detection is reliable, displaying a calibration sensitivity of 0.00202 µF·µM^−1^ (Appendix A), with an LOD of 45.4 µM. CVs acquired using the CNT/CNC sensor electrode during H_2_O_2_ production from the various patches and the resulting plots of the time trend of ∆capacitance are shown in Appendix A, respectively. No H_2_O_2_ was produced via the CNT/CNC@textile patch, evident by the poor linear signal response (R^2^ = 0.0564) (Appendix A). H_2_O_2_ production rates determined by capacitive quantification were 4.19 and 7.20 µM·min^−1^ for the PANI@CNT/CNC@textile and AQ@PANI@CNT/CNC@textile patches, respectively (Figure 5a). Additionally, the total concentration of H_2_O_2_ generated from the PANI@CNT/CNC@textile and AQ@PANI@CNT/CNC@textile patches after 32 min of CV application was 173 and 209 µM, respectively (Figure 5a).

A colorimetric assay was used as an additional method to quantify H_2_O_2_ production [18]. The presence of H_2_O_2_ was indicated by the absorbance peak at 505 nm, and this detection method had a calibration sensitivity of 4.53 × 10^−4^ µM^−1^ (Appendix A), with a LOD of 59.5 µM. The resulting absorption spectra and timed trends in absorbance (@ 505 nm) for H_2_O_2_ generation from AQ@PANI@CNT/CNC@textile and PANI@CNT/CNC@textile patches are shown in Appendix A, respectively. Using the colorimetric assay, the PANI@CNT/CNC@textile patch produced H_2_O_2_ at a rate of 1.01 µM·min^−1^ (Figure 5b). For the AQ@PANI@CNT/CNC@textile patch, H_2_O_2_ production occurred very slowly in the first 16 min before increasing linearly at a rate of 9.54 µM·min^−1^. The total concentration of H_2_O_2_ after 32 min of CV application was determined to be 36.8 and 143 µM for PANI@CNT/CNC@textile and AQ@PANI@CNT/CNC@textile patches, respectively (Figure 5b). Both quantification methods indicate that the integration of AQ into PANI@CNT/CNC@textile patches improves both the rate of and total H_2_O_2_ production (Figure 5). However, capacitive detection is likely more accurate due to a higher calibration sensitivity.

Mechanistically, H_2_O_2_ is generated from O_2_ via a 2e^−^ ORR assisted by the AQ@PANI@CNT/CNC@textile antibacterial patch [1]. AQ acts as the primary catalyst for the OOR reaction, while PANI acts as a conductive surface for the attachment of AQ and contributes additional catalytic activity [1]. Therefore, the greater rates of H_2_O_2_ production for the AQ@PANI@CNT/CNC@textile antibacterial patches relative to the PANI@CNT/CNC@textile can be attributed to the enhanced catalytic activity imparted by the AQ layer [1]. Generally, the AQ@PANI@CNT/CNC@textile patch produces either lower or similar amounts of H_2_O_2_ compared to other methods (Table 2). However, the AQ@PANI@CNT/CNC@textile patch is flexible, lends well for wearability, and is an ideal platform for integration into medical PPE for self-disinfection via on-demand electrochemical H_2_O_2_ generation.

### 3.4. Antibacterial Efficacy of AQ@PANI@CNT/CNC@Textile Antibacterial Patch

The ability of the AQ@PANI@CNT/CNC@textile antibacterial patch to deactivate *E. coli* bacteria was tested. CV was performed on *E. coli*-integrated CNT/CNC@textile, PANI@CNT/CNC@textile, and AQ@PANI@CNT/CNC@textile patches immersed in phosphate buffer, and samples taken from the electrochemical cells at various times were grown on LB agar plates and counted (Appendix A). The CNT/CNC@textile patch showed very little antibacterial activity, demonstrated by the large number of *E. coli* colonies formed from it at any point during CV application (Appendix A). Antibacterial activities of PANI@CNT/CNC@textile and AQ@PANI@CNT/CNC@textile patches were evaluated numerically using a log reduction (Figure 6), calculated by taking the log of the colony forming units (CFU) after 0 min of CV application divided by the CFU after a given time of CV application. After 2 h of CV application, the *E. coli*-integrated AQ@PANI@CNT/CNC@textile antibacterial patch, and the log reduction was 3.32, whereas that of the PANI@CNT/CNC patch was less than 0.500 (Figure 6). As such, the FDA Tier 3 standard for reusable face masks (log reduction ≥ 3) [8] is satisfied via the AQ@PANI@CNT/CNC@textile antibacterial patch, and its antibacterial activity correlates with an approximate 99.9% reduction in *E. coli* bacteria over 2 h. This suggests that the AQ@PANI@CNT/CNC@textile patch may be useful for integration into face coverings or masks for enhanced microbe deactivation and sanitization [28].

## 4. Conclusions

The AQ@PANI@CNT/CNC@textile antibacterial patch described in this study produced H_2_O_2_ in amounts up to 209 ± 25 µM. Although this concentration of H_2_O_2_ is lower than that reported in other studies conducted using PANI and AQ electrocatalysts [12], bacterial deactivation tests showed a log reduction of 3.32 for *E. coli* after 2 h of electrochemical H_2_O_2_ generation. These results are indicative of strong antibacterial activity, and the textile-based antibacterial patch has great potential to be integrated into PPE or face coverings for enhanced protection against pathogens. Overall, this study outlines a promising textile-based technology that may be further optimized for controlling various pathogens in food-safety- and health-related applications.

## Figures and Tables

**Figure 1 polymers-15-02859-f001:**
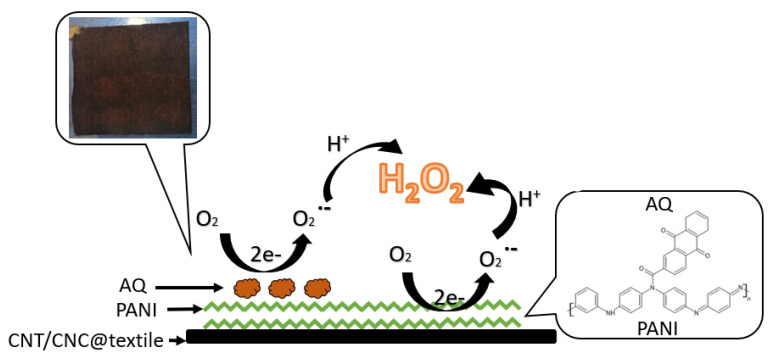
Reaction scheme for AQ- and PANI-catalyzed H_2_O_2_ production.

**Figure 2 polymers-15-02859-f002:**
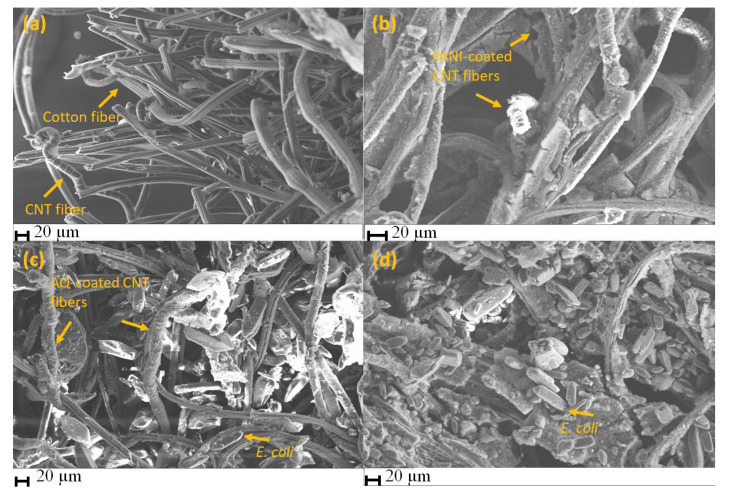
SEM images for (**a**) CNT/CNC@textile, (**b**) PANI@CNT/CNC@textile, (**c**) AQ@CNT/CNC@textile patch after *E. coli* incorporation, and (**d**) AQ@PANI@CNT/CNC@textile patch after *E. coli* incorporation.

**Figure 3 polymers-15-02859-f003:**
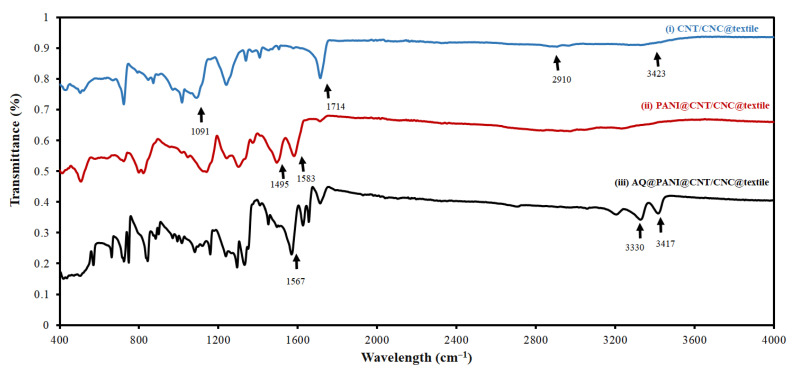
Overlapped FTIR spectra of (**i**) CNT/CNC@textile, (**ii**) PANI@CNT/CNC@textile, and (**iii**) AQ@PANI@CNT/CNC@textile patches.

**Figure 4 polymers-15-02859-f004:**
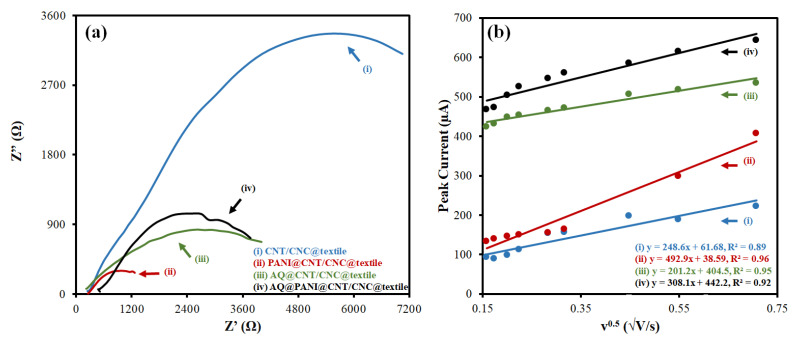
(**a**) Nyquist plots and (**b**) peak current vs. the square root of CV scan rate plots for (**i**) CNT/CNC@textile, (**ii**) PANI@CNT/CNC@textile, (**iii**) AQ@CNT/CNC@textile, and (**iv**) AQ@PANI@CNT/CNC@textile patches.

**Figure 5 polymers-15-02859-f005:**
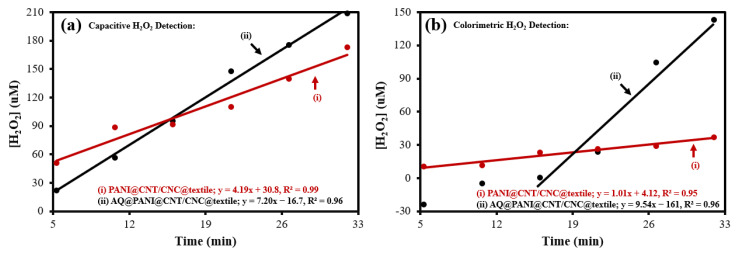
H_2_O_2_ concentration vs. electrochemical H_2_O_2_ generation time plots resulting in linear equations determined from the (**a**) capacitive and (**b**) colorimetric detection methods for (**i**) PANI@CNT/CNC@textile and (**ii**) AQ@PANI@CNT/CNC@textile patches.

**Figure 6 polymers-15-02859-f006:**
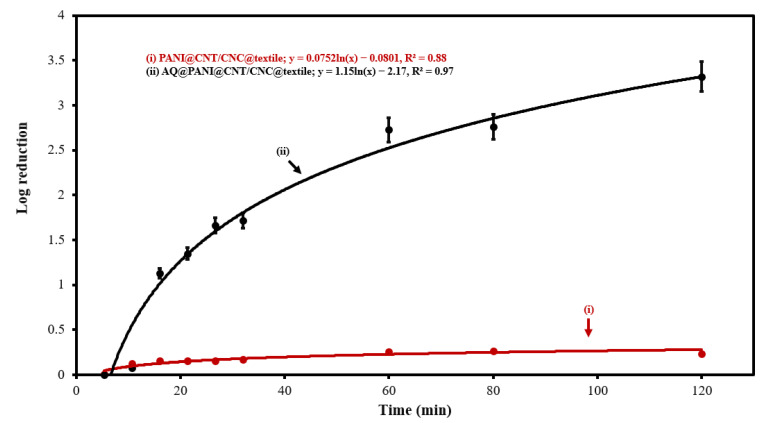
Log reduction of *E. coli* vs. CV application time plot of (**i**) PANI@CNT/CNC@textile and (**ii**) AQ@PANI@CNT/CNC@textile antibacterial patches.

**Table 1 polymers-15-02859-t001:** Electrochemical characteristics of CNT/CNC@textile, PANI@CNT/CNC@textile, AQ@CNT/CNC@textile, and AQ@PANI@CNT/CNC@textile patches.

Patch Type	R_ct_ (Ω)	Electroactive Surface Area (cm^2^)
CNT/CNC@textile	1.63 × 10^4^	13.3
PANI@CNT/CNC@textile	1.45 × 10^3^	24.7
AQ@CNT/CNC@textile	5.13 × 10^3^	10.8
AQ@PANI@CNT/CNC@textile	4.65 × 10^3^	16.5

**Table 2 polymers-15-02859-t002:** Comparison of the rate and total concentration of H_2_O_2_ produced using various methods in the literature.

H_2_O_2_ Production Method	Rate of Production (µM·min^−1^)	Total H_2_O_2_ Produced (µM)	Reference
Electrochemical production using H_2_SO_4_-anodized graphite felt as a cathode	27.1	3.25 × 10^3^	[24]
Electrochemical production using 13-(4-nitrophenyl)-5H-dibenzo[b,i]xanthene-5,7,12,14(13H)-tetraone–modified carbon electrode	113	1.36 × 10^4^	[25]
Photocatalytic production using a catalytic CuBiOS@CuBi_2_O_4_ heterojunction with O-S interpenetration	1.12	202	[26]
Electrochemical production using a carbon-polytetrafluoroethylene-modified carbon cloth gas diffusion electrode	141–329	1.41–3.29 × 10^3^	[27]
Electrochemical production using AQ@PANI@CNT/CNC@textile patch	7.20	209	This work

## Data Availability

All relevant data is included in the manuscript and in the Appendix A.

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
