# Peer review of "Anthraquinone-Polyaniline-Integrated Textile Platforms for In Situ Electrochemical Production of Hydrogen Peroxide for Microbial Deactivation"

_polymers, 2023, doi:10.3390/polym15132859_

Round 1
Reviewer 1 Report (Previous Reviewer 2)
The authors has revised the MS according to the comments.
Author Response
Referee 1
“The authors has revised the MS according to the comments.”

Reviewer 2 Report (Previous Reviewer 1)
The reviewed manuscript presents the application of polyaniline decorated with anthraquinon in the generation of hydrogen peroxide. The polymer was mixed with conductive carbon nanotubes and cellulose nanocrystals and then applied to cotton textiles. The undertaken studies are interesting, and the manuscript may be accepted after minor revision.
Comments:
1. In the text Some typos or unseparated words or phrases may be found, for instance. ofpolyaniline, pathwaythan. Please correct it.
2. In figure 3, it should be wavenumber, not wavelength.
3. Since the polymer was deposited on the surface of textile by means of the layer-by-layer method, could you roughly determine the thickness of this layer?
1. In the text Some typos or unseparated words or phrases may be found, for instance. ofpolyaniline, pathwaythan. Please correct it.
Author Response
Comments:
- In the text Some typos or unseparated words or phrases may be found, for instance. ofpolyaniline, pathwaythan. Please correct it.
We reviewed the manuscript and corrected several of these errors.
- In figure 3, it should be wavenumber, not wavelength.
Corrected.
- Since the polymer was deposited on the surface of textile by means of the layer-by-layer method, could you roughly determine the thickness of this layer?
Each layer deposited onto the textile base is expected to be a few microns thick. SEM images (Figure 2) illustrates this the best; as each textile fiber only slightly visually increasing in thickness as each layer is added.
Comments on the Quality of English Language
- In the text Some typos or unseparated words or phrases may be found, for instance. ofpolyaniline, pathwaythan. Please correct it.
We reviewed the manuscript and corrected several of these errors.
We confirm our manuscript meet all the editorial office comments recommended.
Thank you for reconsidering our revised manuscript.

This manuscript is a resubmission of an earlier submission. The following is a list of the peer review reports and author responses from that submission.
Round 1
Reviewer 1 Report
The presented paper describes Electrochemical generation of hydrogen peroxide by polyaniline (PANI) – anthraquinone based carbon nanotubes/cellulose nanocrystals (CNT/CNC). The obtained textile platform was able to reduce the number of E. coli by 99.9 % over a 2-hour period.
Comments:
1. Please provide a scheme illustrating the mechanism of hydrogen peroxide production using the investigated anthraquinone based material
2. I would recommend to change the word: spectrographs into absorption spectrum.
3. The absorption spectra are too jagged. Please smooth it, at least. Please attribute the color of spectrum to the hydrogen peroxide concentration and show it in the figure S6 and S7.
I recommend this paper for publication in Polymers (MDPI) after minor revision.
Reviewer 2 Report
Comments to polymers-1946386
phosphate buffer was drawn at various points in time (0, 5.3, 10.7, 16, 21.3, 26.7, 32, 60, 80, and 120 min)
Why did you perform the test in time 0, 5.3, 10.7, 16, 21.3, 26.7 min?
To electrochemically generate H2O2, antibacterial patches were immersed in 10 mL of 0.1 M phosphate buffer (pH 7) electrolyte in a three-electrode cell comprising a platinum counter electrode,
To generate H2O2, the patches were immersed into solution. Could this simulate the antimicrobial activity? In solution?
Furthermore, the statistical model is too simple. This could not obtain the conclusion ‘These results are indicative of strong antibacterial activity, and the textile-based antibacterial patch has great potential to be integrated into PPE or face coverings for enhanced protection against pathogens.’
Reviewer 3 Report
I recommend to reject the above mentioned manuscript due to the several reasons:
1. The manuscript is not innovative enough.
2. The clear aim is missing and is not mentioned within Introduction part.
3. Experimental part: majority of reagents and several instruments are of unknown source. According to the guide for authors, manufacturer, type, city and country of the origin has to be given for all instruments as well as supplier, city and country for each reagent.
4. Results part: The explanation of results is very superficial. Moreover, obtained results are introduced but not discussed.
5. The manuscript as it is is not of enough interest for Polymers readers.